# AE-GPT: Using Large Language Models to extract adverse events from surveillance reports-A use case with influenza vaccine adverse events

Yiming Li[1], Jianfu Li[2], Jianping He[1], Cui Tao[2]*

**1** McWilliams School of Biomedical Informatics, The University of Texas Health Science Center at Houston, Houston, TX, United States of America, **2** Department of Artificial Intelligence and Informatics, Mayo Clinic, Jacksonville, FL, United States of America

* Tao.Cui@mayo.edu

## Abstract

Though Vaccines are instrumental in global health, mitigating infectious diseases and pandemic outbreaks, they can occasionally lead to adverse events (AEs). Recently, Large Language Models (LLMs) have shown promise in effectively identifying and cataloging AEs within clinical reports. Utilizing data from the Vaccine Adverse Event Reporting System (VAERS) from 1990 to 2016, this study particularly focuses on AEs to evaluate LLMs' capability for AE extraction. A variety of prevalent LLMs, including GPT-2, GPT-3 variants, GPT-4, and Llama2, were evaluated using Influenza vaccine as a use case. The fine-tuned GPT 3.5 model (AE-GPT) stood out with a 0.704 averaged micro F1 score for strict match and 0.816 for relaxed match. The encouraging performance of the AE-GPT underscores LLMs' potential in processing medical data, indicating a significant stride towards advanced AE detection, thus presumably generalizable to other AE extraction tasks.

## Introduction

Vaccines are a vital component of public health and have been instrumental in preventing infectious illnesses [1, 2]. Nowadays, we possess vaccines that cope with over 20 life-threatening diseases, contributing to enhanced health and longevity for people across all age groups [3]. Each year, vaccinations save between 3.5 to 5 million individuals from fatal diseases such as diphtheria, tetanus, pertussis, influenza, and measles [3]. However, vaccine-related adverse events (AEs), although rare, can occur after immunization. By September 2023, Vaccine Adverse Event Reporting System (VAERS) had received more than 1,791,000 vaccine AE reports, of which 9.5% were classified as serious, which include events that result in death, hospitalization, or significant disability [4–6]. Consequently, vaccine AEs can cause a range of side effects in individuals, from mild, temporary symptoms to severe complications [7–10]. They may also give rise to vaccine hesitancy among healthcare providers and recipients [11].

Understanding AEs following vaccinations is vital in surveilling the effective implementation of immunization programs [12]. Such vigilance ensures the continuous safety of

**Data Availability Statement:** All data underlying the findings described in this manuscript to be freely available at https://www.kaggle.com/ datasets/yimingli99/ae-gpt-data.

**Funding:** This article was partially supported by the National Institute of Allergy And Infectious Diseases of the National Institutes of Health under Award Numbers R01AI130460 and U24AI171008. The funders had no role in study design, data collection and analysis, decision to publish, or preparation of the manuscript.

**Competing interests:** The authors have declared that no competing interests exist.

**Abbreviations:** AE, adverse event; AI, artificial intelligence; CDC, Centers for Disease Control and Prevention; FDA, Food and Drug Administration; GBS, Guillain-Barre syndrome; GPT, Generative Pre-trained Transformer; ICU, intensive care unit; LLM, Large Language Model; NER, named entity recognition; NLP, natural language processing; VAERS, Vaccine Adverse Event Reporting System.

vaccination campaigns, allowing for prompt responses when AE occurs. This not only conduces to recognizing early warning signs but also fosters hypotheses regarding potential new vaccine AEs or shifts in the frequency of known ones, ultimately contributing to the refinement and development of vaccines [13].

Therefore, the extraction of AEs and AE-related information would play a pivotal role in advancing our understanding of conditions such as syndromes and other system disorders that can emerge after vaccination. VAERS functions as a spontaneous reporting system for adverse events post-vaccination, serving as the national early warning mechanism to flag potential safety issues with U.S. licensed vaccines [14–16]. VAERS collects structured information such as age, medical background, and vaccine type. It also includes a short narrative from the reporters to describe symptoms, medical history, diagnoses, treatments, and their temporal information [14]. Although it does not establish causal relationships between AEs and vaccines, VAERS detects possible safety concerns warranting deeper investigation through robust systems and study designs [14]. Du et al. employed advanced deep learning algorithms to detect nervous system disorder-related events in the cases of Guillain-Barre syndrome (GBS) linked to influenza vaccines from the VAERS reports [17]. Through the evaluation of different machine learning and deep learning methods, including domain-specific BERT models such as BioBERT and VAERS BERT, their research demonstrated the superior performance of deep learning techniques over traditional machine learning methods (ie, conditional random fields with extensive features) [17].

Nowadays, with the popularity of artificial intelligence (AI) surging, a remarkable breakthrough has emerged: the development of large language models (LLMs) [18–20]. These cutting-edge AI constructs have redefined the way computers understand and generate human language, leading to unprecedented advancements in various aspects [21]. These models, powered by advanced machine learning techniques, have the capacity to comprehend context, semantics, and nuances, allowing them to generate coherent and contextually relevant text [19]. For example, Generative Pre-trained Transformer (GPT), developed by OpenAI, represents a pioneering milestone in the realm of AI [22]. Built on a massive dataset, GPT is a state-of-the-art language model that excels in generating coherent and contextually relevant text [23]. Since its inception, GPT has exhibited exceptional capabilities, ranging from crafting imaginative narratives and aiding content creation to facilitating language translation and engaging in natural conversations with virtual assistants [24, 25]. Its impact across various domains has highlighted its potential to revolutionize human-computer interaction, making significant strides towards machines truly understanding and interacting with human language [26]. Another model Llama 2, available free of charge, however, takes a novel approach by incorporating multimodal learning, seamlessly fusing text and image data [27]. This unique feature enables Llama 2 to not only comprehend and generate text with finesse but also to understand and contextualize visual information, setting it apart from traditional language models like GPT [27].

LLMs represent a significant leap forward in natural language processing (NLP), enabling applications ranging from text generation and translation to sentiment analysis and Chatbot virtual assistants [28]. By learning patterns from vast amounts of text data, these LLMs have the potential to bridge the gap between human communication and machine understanding, opening up new avenues for communication, information extraction, and problem-solving [29]. Hu et al. examined ChatGPT's potential for clinical named entity recognition (NER) in a zero-shot context, comparing its performance with GPT-3 and BioClinicalBERT on synthetic clinical notes [30]. ChatGPT outperformed GPT-3, although BioClinicalBERT still performed better. The study demonstrates ChatGPT's promising utility for zero-shot clinical NER tasks without requiring annotation [30].

In this study, we aim to develop AE-GPT, an automatic adverse event extraction tool based on LLMs, with a specific focus on adverse events following the Influenza vaccine. The influenza vaccine, being one of the most frequently reported vaccines in the VAERS, serves as a prominent use case for our investigation. Our choice is not only motivated by the vaccine's substantial reporting frequency but also by its significance in public health. As we delve into extracting AE-related entities, the influenza vaccine provides a robust and relevant context for evaluating the performance of our proposed AE-GPT framework. While our study centers on the influenza vaccine as a use case, the framework's applicability extends to other vaccine types, enriching the generalizability of our findings across the broader domain of vaccine safety surveillance. Unlike several studies that have explored the application of LLMs in executing NER tasks, our focus extends beyond zero-shot learning (inference from the pre-trained model), providing comprehensive performance comparisons between pretrained LLMs, fine-tuned LLMs, and traditional language models. This research aims to address this gap by providing a thorough examination of the LLM's capabilities, specifically focusing on its performance within the NER task and also proposing the advanced fine-tuned models AE-GPT, which specializes in AE related entity extraction. Our investigation not only involves utilizing the pretrained model for inference but also enhancing the LLM's NER performance through fine-tuning with the customized dataset, thus providing a deeper understanding of its potential and effectiveness.

## Materials and methods

Fig 1 presents an overview of the study framework. Our investigation commenced with zero-shot entity recognition, involving the direct input of user prompts into pretrained LLMs (GPT & Llama 2). To achieve a comprehensive assessment of LLMs' effectiveness in clinical entity recognition, our analysis covered a range of LLMs, namely GPT-2, GPT-3, GPT-3.5, GPT-4, and Llama 2. Furthermore, to enhance their performance, we performed fine-tuning on these LLMs using annotated data, followed by utilizing user prompts to facilitate result inference.

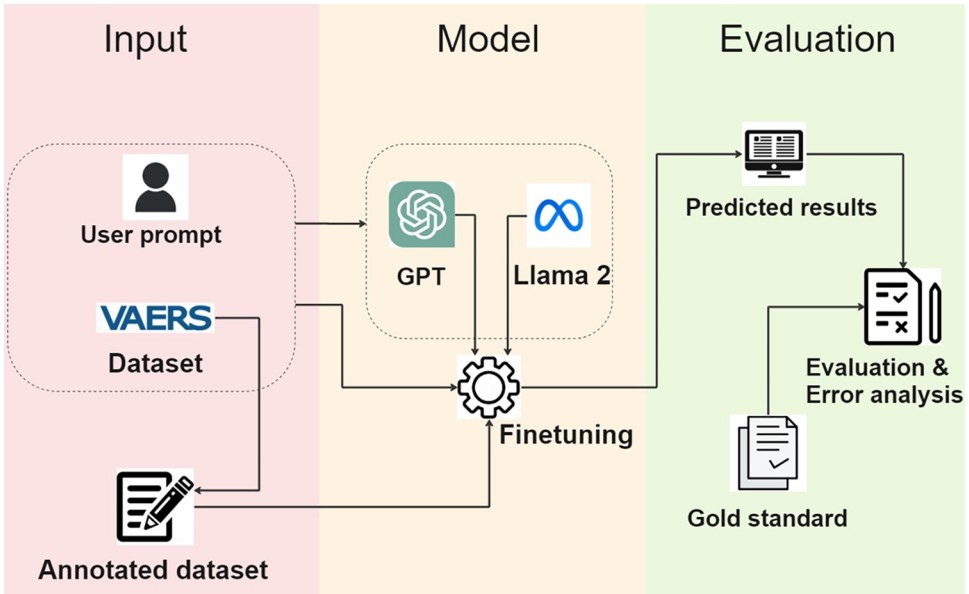

**Fig 1. Overview of the study framework.**

## Data source and use case

VAERS functions as an advanced alert mechanism jointly overseen by the Centers for Disease Control and Prevention (CDC) and the U.S. Food and Drug Administration (FDA), playing a pivotal role in identifying potential safety concerns associated with FDA-approved vaccines [31, 32]. As of Aug 2013, VAERS has documented more than 1,781,000 vaccine-related AEs [32].

The influenza vaccine plays a significant role in preventing millions of illnesses and visits to doctors due to flu-related symptoms each year [33]. For example, in the 2021–2022 flu season, prior to the emergence of the COVID-19 pandemic, flu vaccination was estimated to have prevented around 1.8 million flu-related illnesses, resulting in approximately 1,000,000 fewer medical visits, 22,000 fewer hospitalizations, and nearly 1,000 fewer deaths attributed to influenza [34]. A study conducted in 2021 highlighted that among adults hospitalized with flu, those who had received the flu vaccine had a 26% reduced risk of needing intensive care unit (ICU) admission and a 31% lower risk of flu-related mortality compared to individuals who had not been vaccinated [35].

However, influenza vaccines also have been associated with a range of potential adverse effects, such as pyrexia, hypoesthesia, and even rare conditions like GBS [36]. Among them, GBS ranks as the primary contributor to acute paralysis in developed nations, and continues to be the most frequently documented serious adverse event following trivalent influenza vaccination in the VAERS database, with a report rate of 0.70 cases per 1 million vaccinations [37–39]. This rare autoimmune disorder, GBS, affects the peripheral nervous system, characterized by rapidly advancing, bilateral motor neuron paralysis that typically arises subsequent to an acute respiratory or gastrointestinal infection [37, 40–42].

As a use case, this study focuses on symptom descriptions (referred to as narrative safety reports) that include GBS and symptoms frequently linked with GBS. Particularly, our interest lies in these reports following the administration of diverse influenza virus vaccines, including FLU3, FLU4, H5N1, and H1N1. In order to enable a direct performance comparison with traditional language models, we employed the identical dataset that was utilized by Du et al. in their previous study [17]. This dataset comprises a total of 91 annotated reports. In the context of understanding the development of GBS and other nervous system disorders, we explored six entity types that collectively capture significant clinical insights within VAERS reports: *investigation*, *nervous_AE*, *other_AE*, *procedure*, *social_circumstance*, and *temporal_expression*. *Investigation*, which refers to lab tests and examinations, including entities like "neurological exam" and "lumbar puncture" [17]. *nervous_AE* (e.g., "tremors" "Guillain-Barré syndrome") involves symptoms and diseases related to nervous system disorders, whereas *other_AE* (e.g., "complete bowel incontinence" "diarrhea") is associated with other symptoms and diseases [17]. *procedure* addresses clinical interventions to the patient, including vaccination, treatment and therapy, intensive care, featuring instances such as "flu shot" and "hospitalized" [17]. *social_circumstance* records events associated with the social environment of a patient, for example, "smoking" and "alcohol abuse" [17]. *temporal_expression* is concerned with temporal expressions with prepositions like "for 3 days" and "on Friday morning" [17].

## Models

To fully investigate the performance of LLMs on the NER task, GPT models and Llama 2 will be leveraged.

**GPT.**  GPT represents a groundbreaking advancement in the realm of NLP and artificial intelligence [43]. Developed by OpenAI, GPT stands as a remarkable example of the transformative capabilities of large-scale neural language models [44]. At its core, GPT is founded

upon the innovative Transformer architecture, a model that has revolutionized the field by effectively capturing long-term dependencies within sequences, making it exceptionally well-suited for tasks involving language understanding and generation [45, 46]. The GPT family has multiple versions: The GPT-2 model, with 1.5 billion parameters, is capable of generating extensive sequences of text while adapting to the style and content of arbitrary inputs [47]. Moreover, GPT-2 can also perform various NLP tasks, such as classification [47]. On the other hand, GPT-3 with 175 billion parameters, takes the capabilities even further [35]. It's an auto-regressive language model trained with 96 layers on a combination of 560GB+ web corpora, internet-based book corpora, and Wikipedia datasets, each weighted differently in the training mix [48, 49]. GPT-3 model is available in four versions: Davinci, Curie, Babbage and Ada which differ in the amount of trainable parameters– 175, 13, 6.7 and 2.7 billion respectively [48, 50]. GPT-4 has grown in size by a factor of 1000, now reaching a magnitude of 170 trillion parameters, a substantial increase when compared to GPT-3.5's 175 billion parameters [51]. One of the most notable improvements in GPT-4 is the expanded context length. In GPT-3.5, the context length is 2048 [51]. However, in GPT-4, it has been elevated to 8192 or 32768, depending on the specific version, representing an augmentation of 4 to 16 times compared to GPT-3.5 [51]. In terms of its generated output, GPT-4 possesses the capacity to not just accommodate multimodal input, but also produce a maximum of 24000 words (equivalent to 48 pages) [51]. This represents an increase of 8 times compared to GPT-3.5, constrained by 3000 words (equivalent to 6 pages) [51].

The rationale behind GPT's design stems from the understanding that pre-training, involving the unsupervised learning of language patterns from vast textual corpora, can provide a strong foundation for subsequent fine-tuning on specific tasks [44]. This enables GPT to acquire a sophisticated understanding of grammar, syntax, semantics, and even world knowledge, essentially learning to generate coherent and contextually relevant text [52].

GPT's architecture comprises multiple layers of self-attention mechanisms, which allow the model to weigh the importance of different words in a sentence based on their contextual relationships [53, 54]. This intricate layering, coupled with the model's considerable parameters, empowers GPT to process and generate complex linguistic structures, making it a versatile tool for a wide range of NLP tasks, including text completion, translation, summarization, and even creative writing [55].

**LLama 2.** Llama 2 emerges as a cutting-edge advancement in the domain of natural language processing, marking a significant evolution in the landscape of language models [27]. Developed as an extension of its predecessor, Llama, this model represents an innovative step forward in harnessing the power of transformers for language understanding and generation [56]. The architecture of Llama 2 is firmly rooted in the Transformer framework, which has revolutionized the field by enabling the modeling of complex dependencies in sequences.

The rationale behind Llama 2's conception rests upon the recognition that while pre-training large language models on diverse text corpora is beneficial, customizing their multi-layer self-attention architecture for linguistic structures can further optimize their performance [56].

## Experiment setup

**Dataset split.** In this study, we partitioned the dataset into a training set and a test set using an 8:2 ratio, where 72 VAERS reports were designated for the training set and the remaining 19 reports for the test set.

**Pretrained model inference.** We firstly inferred the results by using the available pre-trained LLMs. GPT-2 model source has been made publicly available, we used

**Table 1. Prompts and hyperparameters of pretrained models.**

| Model | | Prompt | Temperature | Max tokens |
|---|---|---|---|---|
| GPT-2 | | Please extract all names of *investigation*, *nervous AE*, *other AE*, *procedure*, *social circumstance*, and *timestamp* from this note, and put them in a list | 1.0 | 1,000 |
| GPT-3 | ada | Answer the question based on the context below, and if the question can't be answered based on the context, say "I don't know" | 0 | 150 |
| | babbage | Context: [note] | | |
| | curie | — | | |
| | davinci | Question: Please extract all names of [*timestamp/nervous_AE/other_AE/procedure/investigation/social circumstance*] from this note<br>Answer: | | |
| GPT-3.5 | | Please extract all names of *investigation*, *nervous AE*, *other AE*, *procedure*, *social circumstance*, and *timestamp* from this note, and put them in a list | 0.8 | 1,000 |
| GPT-4 | | | | |
| Llama | 2-7b-chat | "Please extract all names of [*timestamp/nervous AE/other AE/procedure/investigation/social circumstance*] from this note: [note] | 0.6 | 512 |
| | 2-13b-chat | | | |

TFGPT2LMHeadModel as the pretrained GPT-2 model to test its ability in this NER task [57]. Llama 2 is also an open-source LLM, which can be accessed through MetaAI [56].

To evaluate the performance of the pre-trained models, we conducted several experiments, selecting the temperature and max tokens settings (shown in Table 1) that yielded the best results. Temperature, a hyperparameter, influences the randomness of the generated text. Higher values, such as 1.0 or above, increase diversity, while lower values, like 0.5 or below, produce more focused outputs. Meanwhile, max tokens determine the maximum length of the generated text, serving to control and limit the length of the output. We employed the prompts (as depicted in Table 1) that adeptly articulate our objective, are comprehensible to the LLMs, and additionally aid in the efficient extraction of results.

Inference for the pretrained GPT models was executed on a server equipped with 8 Nvidia A100 GPUs, where each GPU provided a memory capacity of 80GB. Meanwhile, the pre-trained Llama models were inferred on a server, which included 5 Nvidia V100 GPUs, each offering a memory capacity of 32GB.

**Model fine-tuning.** Fine-tuning the GPT models is facilitated through OpenAI ChatGPT's API calls, with the exception that the GPT-2 model's fine-tuning stems from GPT2sQA and the fine-tuning for GPT-4 has not been made accessible yet [58]. For Llama 2 models, the fine-tuning process begins with HuggingFace. Subsequently, the model's embeddings are automatically fine-tuned and updated. Throughout the process, the temperature remains consistent. The format requirements for training set templates differ among models, depending on whether they are instruction-based or not. Fig 2 presents an example of the question answering-based training set used by GPT-2, which initializes with the question "Please extract all the names of *nervous_AE* from the following note". The question is followed by the answer with annotations where the entities (i.e., Guillain Barre Syndrome, quadriplegic, GBS) and the starting character offset (i.e., 0, 141, 212) should be indicated. The training set ends with the context (Guillain Barre Syndrome. Onset on. . .). Fig 3 shows an example to illustrate the structured format of the training set tailored for GPT-3, where the prompt and annotations are necessitated. In the prompts, only the original report is required because of the predetermined NER template embedded in GPT-3, while the annotations include the entity types and the entities. For instance, as shown in Fig 3, "Guillain Barre Syndrome. Onset. . ." is the original description from the VAERS reports. Within the annotations section, all the

**Fig 2. One example of a question answering-based training set.**

involved entity types (*nervous_AE*, *timestamp*, *investigation*, *other_AE*, and *procedure*) are listed, with 'Guillain Barre Syndrome', 'quadriplegic', and 'GBS' being the entities classified under *nervous_AE*. Fig 4 shows an instruction-based training example utilized for GPT-3.5, and Llama 2-chat, which utilizes prompt instructions to guide and refine the model's responses, ensuring more accurate and contextually relevant outputs. The process of human-machine interaction is imitated. In this scenario, three roles are identified: the system, user, and assistant. The system outlines the task to be accomplished by GPT, stipulating—"You are an assistant adept at named entity recognition." Unlike the structured format training set, in addition to the original VAERS reports, users are also required to clarify the task with a specific question—e.g., "Please extract all the *nervous_AE* entities in the following note." The annotations section only include the anticipated responses that users expect GPT to provide.

For model fine-tuning, we selected the initial hyperparameters as outlined in Table 2. Typically, these settings are based on defaults, except for GPT-3, where non-default values outperformed the defaults. As for prompts, they differ slightly due to model-specific training set needs.

Fine-tuning for the pretrained GPT models was executed on a server equipped with 8 Nvidia A100 GPUs, where each GPU provided a memory capacity of 80GB. Meanwhile, the

**Fig 3. One example of the structured format training set.**

**Fig 4. One example of the instruction-based training set.**

pretrained Llama models were fine-tuned on a server, which included 5 Nvidia V100 GPUs, each offering a memory capacity of 32GB.

**Post-processing.** In the post-processing stage, we addressed instances of nested entities, as depicted in Fig 5 ("Muscle strength" vs "Muscle strength decreased"). To effectively handle this, we adopted a strategy wherein entities possessing the longest spans were retained, while the nested entities were excluded from consideration. In the examples illustrated in Fig 5, the *investigation* term "Muscle strength" was eliminated, resulting in *nervous_AE* "Muscle strength decreased" for the final output. This procedure ensured a streamlined and accurate representation of the entities within the given context.

**Evaluation.** The performance of the LLMs was evaluated by metrics, including precision, recall, and F1. These evaluations were conducted under two distinct matching criteria: exact matching, which required identical entity boundaries, and relaxed matching, which took into consideration overlapping entity boundaries.

$$Precision = \frac{True\ positive}{True\ positive + False\ positive}$$

$$Recall = \frac{True\ positive}{True\ positive + False\ negative}$$

**Table 2. Prompts and hyperparameters of model fine-tuning.**

| Model | | Prompt | Training set format | Temperature | Max tokens |
|---|---|---|---|---|---|
| GPT-2 | | Please extract all the names of [*timestamp/nervous_AE/other_AE/procedure/investigation/ social circumstance*] from the following note | Question answering-based | 1.0 | 1,000 |
| GPT-3 | ada | JSON format specified by OpenAI | Structured | 0.8 | 1,000 |
| | babbage | | | | |
| | curie | | | | |
| | davinci | | | | |
| GPT-3.5 | | Please extract all the [*timestamp/nervous_AE/other_AE/procedure/investigation/social circumstance*] in the following note: [note] | Instruction-based | 1.0 | 4,096 |
| Llama | 2-7b-chat | JSON format specified by Llama | Instruction-based | 1.0 | 4,096 |
| | 2-13b-chat | | | | |

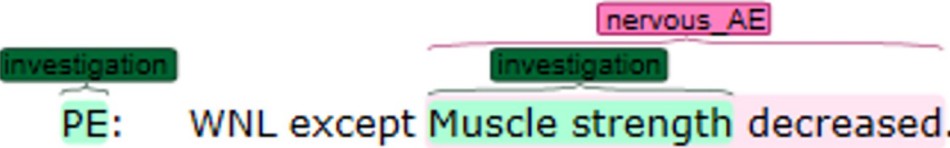

**Fig 5. One example of nested entities.**

$$F - 1 = \frac{2 \times Precision \times Recall}{Precision + Recall}$$

## Results

Tables 3 and 4 presents the NER performance across various LLMs using strict F1 and relaxed F1 metrics respectively. In terms of evaluation metrics, strict F1 requires an exact match in both content and position between the predicted and true segments, while relaxed F1 allows for partial matches, providing a more lenient evaluation of model performance. Notably, the GPT-3.5 model emerges as the frontrunner in this NER task. Remarkably, GPT-3, 3.5, and 4 models surpass LLama models significantly. Within the GPT model family, performance significantly improves with each successive version upgrade. GPT-3-davinci, in particular, achieved the highest performance among all GPT-3 models.

Interestingly, the performance of the fine-tuned GPT-3 model closely rivals that of GPT-4 though the fine-tuned GPT-3 model outperforms both the pretrained GPT-3.5 and GPT-4 models.

Tables 3 and 4 show the NER performance for various entity types, encompassing *investigation*, *nervous_AE*, *other_AE*, *procedure*, *social_circumstance*, and *temporal_expression* respectively. Among these categories, *temporal_expression* exhibits the highest performance, followed by *nervous_AE* and *procedure*.

However, it's worth noting that LLMs encounter significant challenges in extracting social circumstance entities, with the fine-tuned GPT-3.5 model achieving the highest F1 score of 0.5 only in this category. Across all models evaluated, GPT-3.5 generally delivers the best performance, except for the pretrained GPT-3.5, which excels in precision scores for *investigation* extraction. Therefore, we proposed the fine-tuned GPT-3.5, and named it as "AE-GPT".

## Discussion

Our research has yielded remarkable insights into the capabilities of LLMs in the context of NER. With a specific focus on the performance of these models, we have achieved substantial achievements throughout this study. Additionally, we are pleased to introduce the advanced fine-tuned models collectively known as AE-GPT, which have demonstrated exceptional prowess in the extraction of AE related entities. Our work showcases the success of leveraging pretrained LLMs for inference and fine-tuning them with a customized dataset, underscoring the effectiveness and potential of this approach.

In the realm of LLMs for AE NER tasks, the fine-tuned GPT-3.5-turbo model (AE-GPT) notably stood out, demonstrating superior performance compared to its competing models. Interestingly, the process of fine-tuning seemed to have a significant effect on the capabilities of certain models. For instance, both the fine-tuned GPT-3 and GPT-3.5 showed enhanced performance, even outstripping the more advanced but unfine-tuned GPT-4. This suggests

**Table 3. NER performance comparison on VAERS reports by strict F1.**

| | GPT-2 | | GPT-3 ada | | babbage | | curie | | davinci | | GPT-3.5 | | GPT-4 | Llama 2-7b-chat | | 2-13b-chat | |
| --- | --- | --- | --- | --- | --- | --- | --- | --- | --- | --- | --- | --- | --- | --- | --- | --- | --- |
| | Pretrained | Fine-tuned | Pretrained | Fine-tuned | Pretrained | Fine-tuned | Pretrained | Fine-tuned | Pretrained | Fine-tuned | Pretrained | Fine-tuned | Pretrained | Pretrained | Fine-tuned | Pretrained | Fine-tuned |
| investigation | 0 | 0 | 0 | 0.389 | 0 | 0.304 | 0 | 0.214 | 0 | 0.344 | 0.241 | 0.667 | 0.304 | 0.097 | 0.024 | 0.099 | 0.114 |
| nervous_AE | 0 | 0.047 | 0 | 0.319 | 0 | 0.277 | 0 | 0.356 | 0 | 0.459 | 0.208 | 0.727 | 0.458 | 0.102 | 0.024 | 0.173 | 0.096 |
| other_AE | 0 | 0.021 | 0 | 0.17 | 0 | 0.17 | 0 | 0.183 | 0 | 0.351 | 0.165 | 0.638 | 0.412 | 0.234 | 0.041 | 0.213 | 0.042 |
| procedure | 0 | 0.08 | 0 | 0.39 | 0 | 0.388 | 0 | 0.511 | 0 | 0.464 | 0.059 | 0.716 | 0.02 | 0.283 | 0.066 | 0.189 | 0.077 |
| social_circumstance | 0 | 0 | 0 | 0 | 0 | 0 | 0 | 0 | 0 | 0 | 0.133 | 0.5 | 0 | 0 | 0 | 0 | 0 |
| temporal_expression | 0 | 0.075 | 0 | 0.457 | 0 | 0.545 | 0 | 0.504 | 0 | 0.583 | 0.252 | 0.76 | 0.323 | 0.202 | 0.062 | 0.013 | 0.053 |
| Microaverage | 0 | 0.049 | 0 | 0.335 | 0 | 0.356 | 0 | 0.359 | 0 | 0.436 | 0.183 | 0.704 | 0.308 | 0.16 | 0.046 | 0.134 | 0.07 |

Note: The scores were averaged scores after 10 runs

**Table 4. NER performance comparison on VAERS reports by relaxed F1.**

| | GPT-2 | | GPT-3 ada | | babbage | | curie | | davinci | | GPT-3.5 | | GPT-4 | Llama 2-7b-chat | | 2-13b-chat | |
| --- | --- | --- | --- | --- | --- | --- | --- | --- | --- | --- | --- | --- | --- | --- | --- | --- | --- |
| | Pretrained | Fine-tuned | Pretrained | Fine-tuned | Pretrained | Fine-tuned | Pretrained | Fine-tuned | Pretrained | Fine-tuned | Pretrained | Fine-tuned | Pretrained | Pretrained | Fine-tuned | Pretrained | Fine-tuned |
| investigation | 0 | 0 | 0 | 0.463 | 0 | 0.448 | 0 | 0.321 | 0 | 0.53 | 0.289 | 0.795 | 0.464 | 0.13 | 0.047 | 0.128 | 0.21 |
| nervous_AE | 0 | 0.047 | 0 | 0.478 | 0 | 0.408 | 0 | 0.483 | 0 | 0.584 | 0.308 | 0.872 | 0.658 | 0.277 | 0.047 | 0.378 | 0.192 |
| other_AE | 0 | 0.021 | 0 | 0.243 | 0 | 0.243 | 0 | 0.286 | 0 | 0.412 | 0.278 | 0.704 | 0.486 | 0.309 | 0.062 | 0.295 | 0.042 |
| procedure | 0 | 0.08 | 0 | 0.419 | 0 | 0.464 | 0 | 0.534 | 0 | 0.522 | 0.094 | 0.743 | 0.305 | 0.318 | 0.077 | 0.245 | 0.088 |
| social_circumstance | 0 | 0 | 0 | 0 | 0 | 0 | 0 | 0.286 | 0 | 0 | 0.133 | 0.5 | 0 | 0 | 0 | 0 | 0 |
| temporal_expression | 0 | 0.075 | 0 | 0.705 | 0 | 0.743 | 0 | 0.628 | 0 | 0.729 | 0.613 | 0.886 | 0.673 | 0.519 | 0.198 | 0.093 | 0.093 |
| Microaverage | 0 | 0.049 | 0 | 0.457 | 0 | 0.484 | 0 | 0.456 | 0 | 0.54 | 0.329 | 0.816 | 0.515 | 0.269 | 0.101 | 0.221 | 0.118 |

Note: The scores were averaged scores after 10 runs.

that the specific fine-tuning with AE datasets could have equipped GPT models with a more profound insight of the domain, whereas the generic, broad knowledge base of the pretrained GPT-4 may not have been as optimized for this particular task. However, this fine-tuning effect was not universally observed. Despite similar attempts at enhancement, GPT-2 did not exhibit substantial improvements when fine-tuned. One plausible explanation is that GPT-2's underlying architecture and training might have specialized in tasks like text completion rather than NER tasks [59]. Its core strengths may not align as seamlessly with the demands of AE NER, resulting in fine-tuning less effective for this model. On the other hand, the performance of Llama remained stagnant across both its iterations and fine-tuning attempts. This could be indicative of a plateau in the model's learning capacity for the AE NER task, or perhaps the fine-tuning process or data did not sufficiently align with the model's strengths. Another possibility is that Llama's architecture inherently lacks certain features or capacities which make the GPT series more adaptable to the AE NER task. The limited size of the dataset may also contribute to the overfitting, which degrades the performance. Further investigation might be needed to discern the specific factors influencing Llama's performance.

Compared to the work carried out by Du et al., which focused on using the conventional machine learning-based methods and deep learning-based methods [17]. AE-GPT outperforms the proposed model in Du's work (the highest exact match micro averaged F-1 score at 0.6802 by ensembles of BioBERT and VAERS BERT; highest lenient match micro averaged F-1 score at 0.8078 by Large VAERS BERT) [17]. AE-GPT's (the fine-tuned GPT-3.5 model's) enhanced performance in extracting specific entities like investigations, various adverse events, social circumstances, and timestamps can be attributed to its vast pretraining on diverse datasets and its inherent architectural advantages, allowing it to capture broader contextual nuances. Meanwhile, the ensembles of BioBERT and VAERS BERT, despite their biomedical specialization, might have limitations in adaptability across diverse data representations, leading to their comparative underperformance. However, when focusing on *procedure* extraction, the domain-specific nature of the BioBERT and VAERS BERT ensemble might provide a more attuned understanding of the intricate and context-dependent nature of medical procedures. This specificity could overshadow GPT-3.5's broad adaptability, explaining the latter's lesser effectiveness in that particular extraction task.

Our study embarked on a comprehensive comparison of prevalent large language models, encompassing GPT-2, various versions of GPT-3, GPT-3.5, GPT-4, and Llama 2, specifically focusing on their aptitude to extract AE-related entities. Crucially, both pretrained and fine-tuned iterations of these models were scrutinized. Based on its exhaustive nature, this research stands as one of the most holistic inquiries to date into the performance of LLMs in the NER domain. Furthermore, it carves a niche by exploring the impact of fine-tuning on LLMs for NER tasks, distinguishing our efforts from other existing research and reinforcing the study's unique contribution to the field.

While our study offers valuable insights, it is not without its limitations. The dataset utilized in this research is relatively constrained, comprising only 91 VAERS reports. This limited scope might impede the generalizability of our findings to broader contexts. Moreover, it's noteworthy that we primarily focused on VAERS reports, which differ in structure and content from traditional clinical reports, potentially limiting the direct applicability of our findings to other medical documentation.

In our forthcoming endeavors, we aim to incorporate fine-tuning experiments with GPT-4, especially as it becomes accessible for such tasks in the fall of 2023. This will not only add another dimension to our current findings but also ensure that our research remains at the cutting edge, reflecting the latest advancements in the world of LLMs.

## Error analysis

While AE-GPT (the fine-tuned GPT-3.5 model) has demonstrated commendable performance in recognizing a majority of entity types, it exhibits inherent limitations. Table 5 shows the error statistics of AE-GPT across various entity types. Our classification of error types remains consistent with that of Du et al., ensuring easier comparison [17]. 'Boundary mismatch' denotes discrepancies in the span range of entities between machine-annotated and human-annotated results. 'False positive' refers to entities identified by the proposed model that aren't present in the gold standard, while 'false negative' indicates entities the model failed to extract. 'Incorrect entity type' pertains to instances where, though the entity's span range is accurate, the entity itself has been misclassified. It is evident that the model exhibits a predominant challenge in dealing with boundary mismatch, false positives and false negatives, which can be attributed to several factors. The quality and representativeness of the training data play a significant role; inconsistent or limited annotations can lead to mismatches and incorrect identifications [60, 61]. The inherent complexity of distinguishing similar and potentially overlapping entities adds to the challenge. Additionally, textual ambiguity and the trade-off between specialization during fine-tuning and the generalization from the model's original vast pretraining can impact accuracy. While GPT-3 is powerful, capturing all the nuances of a specialized NER task can still pose challenges.

In particular, AE-GPT tends to miss specific *procedure* names, such as "IV immunoglobulin" and "flu vax." Likewise, it exhibits a heightened likelihood of failing to recognize entities related to social circumstances. This underscores the necessity for an improved and broader domain-specific vocabulary within GPT-3.5.

Moreover, AE-GPT frequently confuses general terms such as "injection" and "vaccinated" as exact *procedure* names, and fails to extract the real vaccine names following it in the text. This misinterpretation results in concurrent false positive and false negative errors.

Another noteworthy limitation of AE-GPT is its proneness to splitting errors. For instance, the named phrase "unable to move his hands, arms or legs.", the model often erroneously segments this into "unable to move his hands" and "arms or legs," revealing a shortcoming in its grasp of language understanding.

In our next steps, we intend to improve rare entity extraction, such as social circumstances, by leveraging ontologies and terminologies in these specific domains. We also plan to enhance the embeddings within LLMs to broaden their coverage of these rare entities. Furthermore, expanding our dataset to include drug AEs is on our agenda. We will also introduce clinical notes and biomedical literature to further enrich the dataset. This increased data volume will enable LLMs to better distinguish nuances between entity classes, such as *procedure* vs. *investigation* and *nervous_AE* vs. *other_AE*.

**Table 5. Statistics of AE-GPT prediction errors on different entity types.**

| | Boundary Mismatch (out of human annotated entities) | False Positive (out of machine annotated entities) | False Negative (out of human annotated entities) | Incorrect Entity Type (out of machine annotated entities) |
|---|---|---|---|---|
| **investigation** | 13/66, 19.7% | 20/90, 22.22% | 6/66, 9.1% | 5/90, 5.56% |
| **nervous_AE** | 28/175, 16% | 15/169, 8.88% | 30/175, 17.14% | 1/169, 0.59% |
| **other_AE** | 12/169, 7.1% | 21/132, 15.91% | 63/169, 37.28% | 3/132, 2.27% |
| **procedure** | 4/156, 2.56% | 28/140, 20% | 46/156, 29.49% | 2/140, 1.43% |
| **social_circumstance** | 0/2, 0% | ½, 50% | ½, 50% | 0/2, 0% |
| **temporal_expression** | 21/141,14.89% | 6/130, 4.62% | 22/141,15.6% | 0/130,0% |

In future investigations, we will also acknowledge the importance of delving into the statistical significance of identified AEs. While the current study primarily focuses on evaluating the performance of different pretrained and fine-tuned LLMs in the NER task, we have previously conducted statistical analyses using structured data [62, 63]. The subsequent phase of assessing the statistical significance of AEs represents a crucial avenue for further exploration. Our plan is to integrate the extracted data from unstructured text with the previously collected structured data, incorporating rigorous statistical methods, such as hypothesis testing and significance thresholds. This approach aims to systematically evaluate the significance of AEs within the context of the Influenza vaccine use case, providing a more comprehensive understanding and enhancing the robustness and reliability of our findings.

## Conclusion

In conclusion, our comprehensive exploration of LLMs within the context of NER, including the development of our specialized AE extraction model AE-GPT, has not only highlighted the profound implications of our findings but also marks a significant achievement as the first paper to evaluate the realm of pretrained and fine-tuned LLMs in NER. The introduction of our specialized fine-tuned model, AE-GPT, exhibits the ability to tailor LLMs to domain-specific tasks, offering promising avenues for addressing real-world challenges, particularly in the extraction of AE related entities. Our research underlines the broader significance of LLMs in advancing natural language understanding and processing, with implications spanning various fields, from healthcare and biomedicine to information retrieval and beyond. As we continue to harness the potential of LLMs and refine their performance, we anticipate further breakthroughs that will drive innovation and enhance the utility of these models across diverse applications.

## Author Contributions

**Data curation:** Yiming Li.

**Formal analysis:** Yiming Li.

**Funding acquisition:** Cui Tao.

**Methodology:** Yiming Li, Jianfu Li, Jianping He, Cui Tao.

**Project administration:** Cui Tao.

**Software:** Yiming Li.

**Supervision:** Cui Tao.

**Validation:** Jianfu Li.

**Visualization:** Yiming Li.

**Writing – original draft:** Yiming Li.

**Writing – review & editing:** Jianfu Li, Cui Tao.

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
