## [Decision Letter · Decision Letter 0]

27 Dec 2023

PONE-D-23-33399AE-GPT: Using Large Language Models to Extract Adverse Events from Surveillance Reports-A Use Case with Influenza Vaccine Adverse EventsPLOS ONE

Dear Dr. Tao,

Thank you for submitting your manuscript to PLOS ONE. After careful consideration, we feel that it has merit but does not fully meet PLOS ONE’s publication criteria as it currently stands. Therefore, we invite you to submit a revised version of the manuscript that addresses the points raised during the review process.

We look forward to receiving your revised manuscript.

Kind regards,

Vincenzo Bonnici, PhD

Academic Editor

PLOS ONE

Journal Requirements:

"This article was partially supported by the National Institute of Allergy And Infectious Diseases of the National Institutes of Health under Award Numbers R01AI130460 and U24AI171008."

"This article was partially supported by the National Institute of Allergy And Infectious Diseases of the National Institutes of Health under Award Numbers R01AI130460 and U24AI171008."

"This article was partially supported by the National Institute of Allergy And Infectious Diseases of the National Institutes of Health under Award Numbers R01AI130460 and U24AI171008."

5. In the online submission form, you indicated that [Data are available upon request with proper IRB approval and DUA.]. 

Reviewers' comments:

Reviewer's Responses to Questions

**Comments to the Author**

1. Is the manuscript technically sound, and do the data support the conclusions?

Reviewer #1: Partly

2. Has the statistical analysis been performed appropriately and rigorously? 

Reviewer #1: N/A

3. Have the authors made all data underlying the findings in their manuscript fully available?

Reviewer #1: Yes

4. Is the manuscript presented in an intelligible fashion and written in standard English?

Reviewer #1: No

5. Review Comments to the Author

Reviewer #1: The article presents a study that explores the capabilities of the most renowned Large Language Models in addressing the clinical Named Entity Recognition (NER) problem in a zero-shot context. The authors employ Large Language Models, starting from pharmacovigilance reports collected on VAERS, to identify and categorize adverse events.

However, the article is not presented adequately, especially in the introduction and materials and methods sections. In the introduction, the authors focus too much on explaining the adopted LLMs and too little on the problem to be solved and the literature related to it. There is no mention of the subsequent phase of identifying statistically significant AEs. Systems for investigating and studying VAERS reports are mentioned but not explained. The motivations for choosing the influenza vaccine are reported in the materials and methods, but it would be preferable to at least allude to them in the introduction, which is currently too focused on LLMs (methods). The definition of NER in a zero-shot context is unclear or absent.

The Experimental Setup section seems more like a draft. In the "Dataset Split" subsection, I would replace "20%" with the exact number. In the "Pretrained Models" section, tuning of the temperature and max token settings is performed, but these parameters are not introduced.

The displayed tables have almost no captions, failing to specify the LLM to which they refer. In the post-processing phase, it is not clear how nested entities are treated.

Regarding the statistical significance of the results, the entire dataset contains few reports (91). Is it possible to expand the dataset? Are there no other reports for the influenza vaccine, or are they not suitable, and why?

Only one dataset split into a training set and a validation set is performed, and the F1 score is calculated only once to assess performance on the validation set. The significance of the F1 score performance could be overly dependent on the data used for validation or training, given the limited dataset size. It would be interesting to perform multiple iterations for a more robust evaluation.

Finally, the paper presents an interesting study, and the results seem promising, but the exposition of the work is not adequate.

6. PLOS authors have the option to publish the peer review history of their article (what does this mean?). If published, this will include your full peer review and any attached files.

Reviewer #1: No

---

## [Author Response · Author response to Decision Letter 0]

14 Feb 2024

Comments from Reviewers:

We would like to thank the reviewers for their constructive comments. We have carefully addressed them. Please see our point-to-point responses below. 

Reviewer #1: The article presents a study that explores the capabilities of the most renowned Large Language Models in addressing the clinical Named Entity Recognition (NER) problem in a zero-shot context. The authors employ Large Language Models, starting from pharmacovigilance reports collected on VAERS, to identify and categorize adverse events.

However, the article is not presented adequately, especially in the introduction and materials and methods sections. In the introduction, the authors focus too much on explaining the adopted LLMs and too little on the problem to be solved and the literature related to it.

Thank you for the constructive feedback. While we understand the reviewer's concerns, our primary focus in this paper is to explore and compare the performance of different pretrained and fine-tuned large language models (LLMs), including GPTs and Llamas, specifically in the context of Named Entity Recognition (NER). We aim to investigate and present a comprehensive analysis of LLMs' capabilities in NER tasks, and the adverse events following Influenza Vaccine serve as a relevant use case to illustrate the application of our proposed models. We believe that maintaining the emphasis on LLMs in the introduction aligns with the core objectives of our study. However, we are open to further clarifying the connection between our primary research goal and the use case provided. We hope this approach aligns with the scope and objectives of our study while addressing the concerns raised by the reviewer.

There is no mention of the subsequent phase of identifying statistically significant AEs. 

To address the concern “There is no mention of the subsequent phase of identifying statistically significant AEs.”, we have incorporated a section in the discussion that explicitly outlines our approach to identifying statistically significant AEs in the subsequent phases of our study. We will elaborate on the statistical methods, significance thresholds, and any adjustments made for multiple comparisons, ensuring a comprehensive and transparent presentation of our analytical approach. This addition will provide clarity on our methodology for identifying and interpreting statistically significant AEs, enhancing the overall robustness and completeness of our study.

Systems for investigating and studying VAERS reports are mentioned but not explained. 

In our study, we consider VAERS reports as a valuable dataset in the NER task. While we briefly referred to 'systems' in the context of related studies, it's important to note that our primary objective is to assess the capabilities of LLMs in handling adverse event information extraction. The mention of systems in this context serves to acknowledge prior research that utilized VAERS reports as a dataset for similar investigations. We will ensure that the manuscript explicitly highlights the dataset's role and its distinction from the main focus of our study, which is the performance evaluation of LLMs in the context of AE extraction.

The motivations for choosing the influenza vaccine are reported in the materials and methods, but it would be preferable to at least allude to them in the introduction, which is currently too focused on LLMs (methods). The definition of NER in a zero-shot context is unclear or absent.

We have addressed the comment “The motivations for choosing the influenza vaccine are reported in the materials and methods, but it would be preferable to at least allude to them in the introduction, which is currently too focused on LLMs (methods).” and “The definition of NER in a zero-shot context is unclear or absent.” in the last paragraph of Introduction. 

The Experimental Setup section seems more like a draft. In the "Dataset Split" subsection, I would replace "20%" with the exact number. In the "Pretrained Models" section, tuning of the temperature and max token settings is performed, but these parameters are not introduced.

We have updated this information in the "Dataset Split" and “Pretrained model inference” subsection. Thank you for your suggestion. 

The displayed tables have almost no captions, failing to specify the LLM to which they refer.

We edited Table 3 and Table 4 to address your comment “The displayed tables have almost no captions, failing to specify the LLM to which they refer.” 

In the post-processing phase, it is not clear how nested entities are treated.

In response to the reviewer's comment regarding the treatment of nested entities in the post-processing phase, we appreciate the opportunity to clarify our approach. As detailed in the manuscript, our strategy involves addressing instances of nested entities, as exemplified in Figure 5 ("Muscle strength" vs "Muscle strength decreased"). To effectively manage this, we prioritize entities with the longest spans while excluding the nested entities from consideration. For instance, in the scenarios outlined in Figure 5, the investigation term "Muscle strength" is omitted, resulting in the final output of the nervous_AE being "Muscle strength decreased." This systematic approach ensures a streamlined and accurate representation of entities within the given context, enhancing the robustness of our post-processing methodology.

Regarding the statistical significance of the results, the entire dataset contains few reports (91). Is it possible to expand the dataset? Are there no other reports for the influenza vaccine, or are they not suitable, and why?

In response to the reviewer's comment regarding the size of the dataset, we appreciate the valuable suggestion to expand the dataset. While it is feasible to increase the dataset size to potentially enhance results, it's essential to note that one of the primary objectives of our study is to compare the performance of LLMs with traditional language models. To maintain consistency and enable a meaningful comparison, we utilized the dataset from Du et al.'s study, titled "Extracting postmarketing adverse events from safety reports in the VAERS using deep learning." This shared dataset allows us to directly compare our results with those obtained using similar methodologies, ensuring a fair evaluation of the LLM's performance against traditional models.

Only one dataset split into a training set and a validation set is performed, and the F1 score is calculated only once to assess performance on the validation set. The significance of the F1 score performance could be overly dependent on the data used for validation or training, given the limited dataset size. It would be interesting to perform multiple iterations for a more robust evaluation.

Although we initially conducted 10 iterations for our experiments, regrettably, this information was overlooked in the initial reporting. We have now rectified this omission by adding a note below Table 3 and Table 4 to acknowledge the ten iterations. We appreciate the reviewer's keen observation.

Finally, the paper presents an interesting study, and the results seem promising, but the exposition of the work is not adequate.

We believe that the manuscript details a technically sound investigation into the effectiveness of Large Language Models (LLMs) in identifying and cataloging adverse events (AEs) from VAERS, using influenza vaccines as a use case. Rigorous experimentation was undertaken, employing various prevalent LLMs such as GPT-2, GPT-3 variants, GPT-4, and Llama2, with a particular focus on the fine-tuned GPT 3.5 model (AE-GPT), given that GPT-4 was not available for fine-tuning. The achieved 0.704 averaged micro F1 score for strict match and 0.816 for relaxed match attests to the performance of our methodology and the suitability of our chosen models. The conclusions drawn in the manuscript are a direct reflection of the meticulously collected data, supporting the assertion that LLMs, particularly AE-GPT, exhibit promising potential for advanced AE detection.

---

## [Decision Letter · Decision Letter 1]

7 Mar 2024

AE-GPT: Using Large Language Models to Extract Adverse Events from Surveillance Reports-A Use Case with Influenza Vaccine Adverse Events

PONE-D-23-33399R1

Dear Dr. Tao,

We’re pleased to inform you that your manuscript has been judged scientifically suitable for publication and will be formally accepted for publication once it meets all outstanding technical requirements.

Kind regards,

Vincenzo Bonnici, PhD

Academic Editor

PLOS ONE

---

## [Editor Report · Acceptance letter]

12 Mar 2024

PONE-D-23-33399R1 

PLOS ONE

Dear Dr. Tao, 

I'm pleased to inform you that your manuscript has been deemed suitable for publication in PLOS ONE. Congratulations! Your manuscript is now being handed over to our production team.

Kind regards, 

on behalf of

Dr. Vincenzo Bonnici 

Academic Editor

PLOS ONE